# Cannabinoids modulate associative cerebellar learning via alterations in behavioral state

**Catarina Albergaria, N Tatiana Silva, Dana M Darmohray, Megan R Carey\***

Champalimaud Neuroscience Program, Champalimaud Center for the Unknown, Lisbon, Portugal

**Abstract** Cannabinoids are notorious and profound modulators of behavioral state. In the brain, endocannabinoids act via Type 1-cannabinoid receptors (CB1) to modulate synaptic transmission and mediate multiple forms of synaptic plasticity. CB1 knockout (CB1KO) mice display a range of behavioral phenotypes, in particular hypoactivity and various deficits in learning and memory, including cerebellum-dependent delay eyeblink conditioning. Here we find that the apparent effects of CB1 deletion on cerebellar learning are not due to direct effects on CB1-dependent plasticity, but rather, arise as a secondary consequence of altered behavioral state. Hypoactivity of CB1KO mice accounts for their impaired eyeblink conditioning across both animals and trials. Moreover, learning in these mutants is rescued by walking on a motorized treadmill during training. Finally, cerebellar granule-cell-specific CB1KOs exhibit normal eyeblink conditioning, and both global and granule-cell-specific CB1KOs display normal cerebellum-dependent locomotor coordination and learning. These findings highlight the modulation of behavioral state as a powerful independent means through which individual genes contribute to complex behaviors.

**\*For correspondence:**
megan.carey@neuro.
fchampalimaud.org

## Introduction

Signals relating to ongoing behavior are widely represented in the brain (*Musall et al., 2019*; *Stringer et al., 2019*; *Powell et al., 2015*). Sensorimotor signals can be used to monitor and refine ongoing movements, while generalized changes in behavioral state, including arousal and levels of locomotor activity, influence sensory processing and perception (*Niell and Stryker, 2010*; *Ayaz et al., 2013*; *McGinley et al., 2015*; *Schneider and Mooney, 2015*; *Vinck et al., 2015*; *Pakan et al., 2016*). Both locomotor activity and arousal modulate delay eyeblink conditioning, a form of cerebellum-dependent associative learning (*Albergaria et al., 2018*).

Neuromodulatory systems often directly affect behavioral state and also modulate synaptic transmission and plasticity. For example, cannabinoids are profound modulators of behavioral state, across species (*Mackie, 2007*; *Oakes et al., 2017*; *Luchtenburg et al., 2019*). Throughout the brain, endocannabinoids act as retrograde messengers to regulate neurotransmitter release, mainly via Type-1 cannabinoid (CB1) receptors. Presynaptic CB1 receptors also mediate short- and long-term forms of synaptic plasticity in a wide variety of brain regions and cell types, raising the possibility that they provide a direct substrate for learning and memory (*Freund et al., 2003*; *Mackie, 2006*; *Chevaleyre et al., 2006*; *Regehr et al., 2009*).

CB1 knockout mice (CB1KO) display a variety of behavioral phenotypes (*Zimmer et al., 1999*; *Valverde et al., 2005*). They exhibit altered feeding behavior, anxiety, and nociception (*Zimmer et al., 1999*; *Degroot and Nomikos, 2004*; *Ravinet Trillou et al., 2004*). Notably, global CB1KOs are hypoactive, and spend less time running than their wildtype littermates (*Zimmer et al., 1999*; *Dubreucq et al., 2010*; *Chaouloff et al., 2011*). This hypoactivity has been attributed to the loss of CB1 receptors from GABAergic neurons in the ventral tegmental area (*Dubreucq et al.,*

2013). They also display a range of impairments in learning and memory, including spatial memory and fear conditioning (*Varvel and Lichtman, 2002*; *Marsicano et al., 2002*; *Pamplona and Takahashi, 2006*).

In the cerebellum, CB1 receptors are highly expressed in granule cells and are required for several forms of plasticity within the cerebellar cortex. They mediate short-term suppression of excitation and inhibition, which act on a timescale of tens of milliseconds to several seconds, as well as multiple forms of long-term plasticity (*Soler-Llavina and Sabatini, 2006*; *Kreitzer and Regehr, 2001b*; *Brenowitz and Regehr, 2005*; *Kreitzer and Regehr, 2001a*; *Brown et al., 2003*; *Diana and Marty, 2004*). In particular, CB1 receptors have been implicated in long-term depression (LTD) between parallel fibers and Purkinje cells, an oft-hypothesized, but widely debated, substrate for cerebellar learning (*Albus, 1971*; *Ito, 1972*; *Schonewille et al., 2011*; *Johansson et al., 2018*). Neither global CB1KOs nor granule-cell-specific CB1KOs express LTD in brain slices (*Safo and Regehr, 2005*; *Carey et al., 2011*).

A previous study found that global CB1KOs were impaired in delay eyeblink conditioning (*Kishimoto and Kano, 2006*), a form of cerebellum-dependent associative learning (*McCormick and Thompson, 1984*; *Kim and Thompson, 1997*; *Heiney et al., 2014*). Pharmacological evidence also implicates CB1 signaling in eyeblink conditioning. Systemic CB1 antagonists impair acquisition of delay eyeblink conditioning in mice and rats (*Kishimoto and Kano, 2006*; *Steinmetz and Freeman, 2010*; *Steinmetz and Freeman, 2020*), as do both systemic (*Steinmetz and Freeman, 2016*) and intracerebellar (*Steinmetz and Freeman, 2020*) CB1 agonists. However, a critical role for cerebellar CB1-mediated plasticity in cerebellar learning has not been directly established.

We recently demonstrated that engaging in locomotor activity enhances delay eyeblink conditioning within the cerebellum (*Albergaria et al., 2018*), raising the question of whether hypoactivity could indirectly contribute to the apparent impairments of CB1KO mice in cerebellar learning. Here we find that decreased locomotor activity fully accounts for the effects of CB1 deletion on eyeblink conditioning. Moreover, both global and cerebellar granule-cell-specific CB1KO mice show normal cerebellum-dependent locomotor learning. We conclude that the previously described effects of CB1R deletion on cerebellar learning arise as a secondary consequence of hypoactivity in CB1KOs, and not from direct effects on cerebellar plasticity. These findings highlight the modulation of behavioral state, including locomotor activity, as a powerful mechanism through which individual genes contribute to cognition and behavior.

## Results

### CB1KOs are hypoactive and have impaired eyeblink conditioning

We tested cerebellum-dependent delay eyeblink conditioning in global CB1KO mice and their wild-type littermates using a head-fixed apparatus with a freely rotating running wheel as previously described (*Figure 1A,B*; *Albergaria et al., 2018*). Conditioning sessions included 100 trials in which a neutral visual conditioned stimulus (CS, a white LED) was paired with an air-puff unconditioned stimulus (US). The CS preceded the US by 300 ms and the two stimuli co-terminated. We measured the mouse's locomotor activity continuously with an infra-red sensor placed underneath the wheel (*Figure 1A*).

Consistent with a previous study that found impaired learning over seven acquisition sessions (*Kishimoto and Kano, 2006*), we found that CB1KOs displayed delayed learning, as measured by the percentage of trials that yielded learned conditioned responses (CRs; *Figure 1C*).

When compared to their littermate controls, CB1KO mice displayed significantly less locomotor activity on the self-paced running wheel during training sessions (*Figure 1D*, p=0.015; *Figure 1—figure supplement 1A*), consistent with the previously described hypoactivity of these mice (*Chaouloff et al., 2011*; *Dubreucq et al., 2010*; *Zimmer et al., 1999*). Similar hypoactivity was observed regardless of whether it was measured as total distance during the session (*Figure 1D*), running speed per trial (*Figure 1—figure supplement 1B*, p=0.013), % of time walking (*Figure 1—figure supplement 1C*, p=0.002), or % of trials with ambulatory activity (*Figure 1—figure supplement 1D*, p=0.007). This generalized decrease in ambulation may reflect decreased arousal, as pupil sizes were also reduced in these mice (*Figure 1—figure supplement 1E*), p=0.008; (*Vinck et al., 2015*; *Mineault et al., 2016*; *Reimer et al., 2014*; *Reimer et al., 2016*).

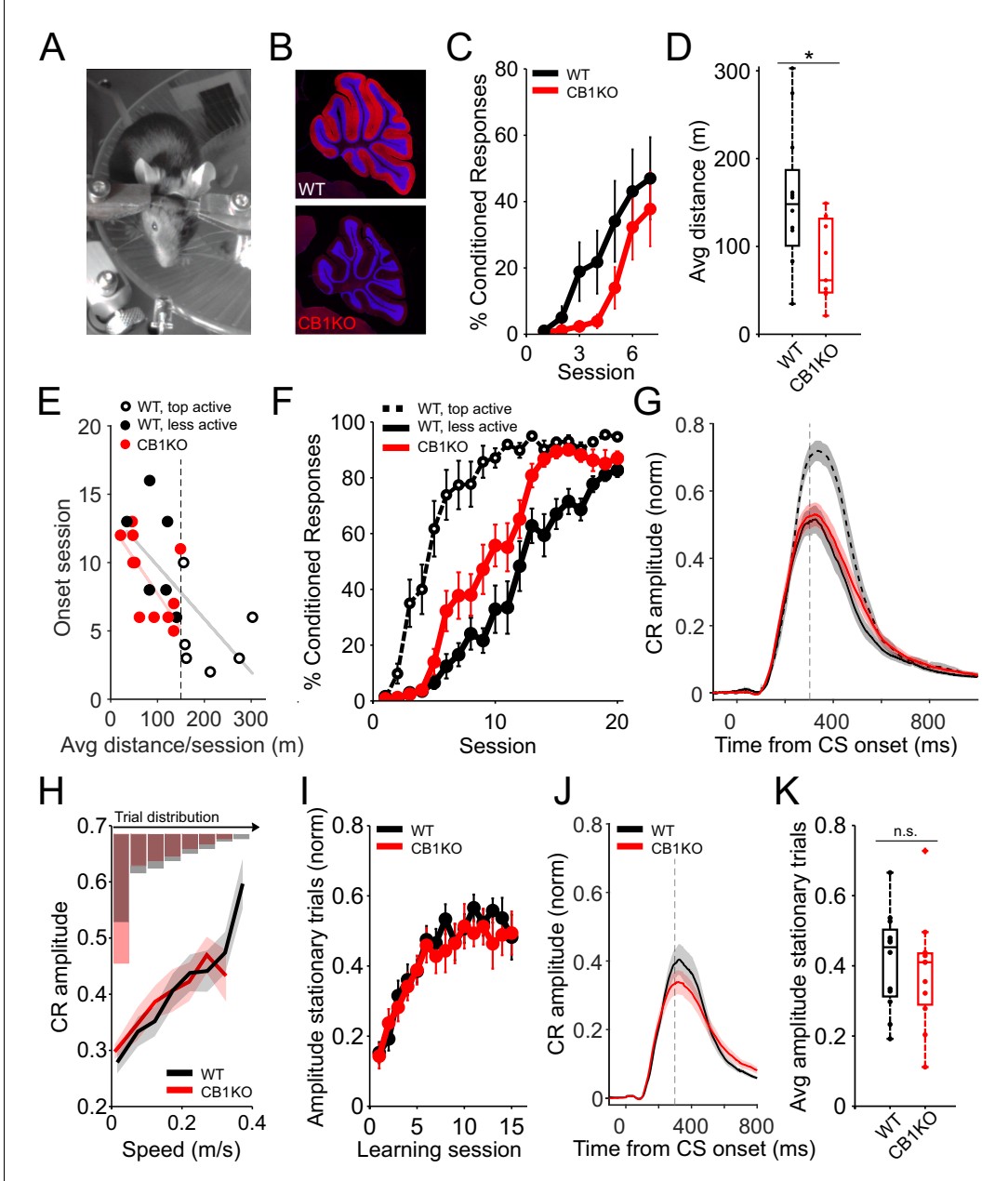

**Figure 1.** Correcting for differences in locomotor activity accounts for apparent deficits in eyeblink conditioning. (A) Setup for eyeblink conditioning in head-fixed mice on a running wheel, showing an LED as the CS and an air-puff US. (B) Sagittal sections showing the complete elimination of CB1R (red) in CB1KOs and DAPI staining the granule-cell layer (blue). Images of the cerebellum of a representative control (upper panel) and global CB1 knockout (bottom panel). (C) Average %CR learning curves of global CB1 knockout mice (CB1KO, red, N = 11) and their littermate controls (WT, black, N = 12). Error bars indicate SEM. (D) Average running distance of WT (black, N = 12) and CB1KO (red, N = 11) on a self-paced treadmill across all learning sessions. Global CB1 knockout mice were significantly hypoactive (*p=0.015). Box indicates median and 25th to 75th percentiles, whiskers extend to the most extreme data points. (E) Onset session of learning for CB1KO (red) and littermate controls (black), divided into top and bottom runners (open and filled circles, respectively), plotted against the animals' mean walking distance, averaged across all 20 training sessions. Vertical dashed line indicates the threshold used for dividing the control animals (150 meters/session, on average). Onset session was defined as the session in which the average CR amplitude exceeded 0.1. Each dot represents an animal. Lines are linear robust fits for CB1KO (red, slope = −0.05, *p=0.02) and controls (gray, slope = −0.04, *p=0.03). (F) Average %CR learning curves of CB1KO (red line, N = 11) compared with wildtype littermates with comparable (solid black line, N = 6) or increased locomotor activity levels (dashed black line, N = 6). Error bars indicate SEM. (G) Average eyelid traces of CS-only trials from the last two training sessions for CB1KO (red line, N = 11), hypoactive wildtype animals (solid black line, N = 6) and littermate controls with increased locomotor activity (dashed black line, N = 6). Vertical dashed line represents the time that the US would have been expected on CS+US trials. Shadows indicate SEM. (H) Trial-to-trial correlation between eyelid response amplitude and walking speed. Amplitudes for all trials from six training sessions
*Figure 1 continued on next page*

*Figure 1 continued*

following learning onset (defined as in (E)) are plotted with lines representing averages across control (black) and global knockout (red) animals; shadows indicate SEM. There was a linear positive relationship for both controls (N = 12, F(1,65.9) = 34.7, ***p=1.5e-07) and global knockouts (N = 11, F(1,43.5) = 43.4, ***p=4.9e-08). Histograms indicate the relative % of trials (averaged across animals) from each genotype that fell in each speed bin. (I) Average amplitude learning curves including only stationary trials of control (WT, black, N = 12) and CB1KO mice (red, N = 11), aligned for each animal's onset session (defined as in (E)). Error bars indicate SEM. (J) Average eyelid traces of stationary (<0.05 m/s) CS-only trials for control (black) and CB1KO (red) animals, from same sessions in (I). Shadows indicate SEM. Vertical dashed line represents the time that the US would have been expected on CS+US trials. (K) Mean eyelid amplitudes from stationary trials from same sessions as in (I). There was no significant difference in the average amplitude of control (black, N = 12) vs. CB1KO (red, N = 11) animals (p=0.58). Dots represent individual animals. Box indicates median and 25th to 75th percentiles, whiskers extend to the most extreme data points.

The online version of this article includes the following source data and figure supplement(s) for figure 1:

**Source data 1.** Source data for *Figure 1* and related supplements.
**Figure supplement 1.** Various measures of altered behavioral state in CB1KO mice and relationship to learning.
**Figure supplement 2.** Learned response timing and short-term, single-trial learning are normal in global CB1KO mice.

## Hypoactivity in CB1KOs accounts for apparent learning deficits

We previously showed that eyeblink conditioning is modulated by behavioral state, and specifically, enhanced by locomotor activity (*Albergaria et al., 2018*). We therefore asked whether the hypoactivity of global CB1KOs could contribute to their delayed learning.

Over the course of 20 training sessions, both controls and CB1KOs eventually learned the task (*Figure 1E,F*). Consistent with our previous study, we found that regardless of genotype, the more an animal ran on average throughout the training trials, the earlier it learned (*Figure 1E*; controls, black circles: slope = −0.13, p=0.04; CB1KO, red circles: slope = −0.12, p=0.02; *Figure 1—figure supplement 1F–G*). Notably, CB1KOs and controls with similar levels of locomotor activity learned at similar rates (*Figure 1E,F*, filled circles). This relationship held for various measures of locomotor activity (*Figure 1—figure supplement 1I–L*). Comparing acquisition curves of CB1KOs with comparably active littermate controls (*Figure 1F*, filled circles) also revealed no delays (and if anything, a relative enhancement) in learning in the CB1KOs. These results suggest that the slower average learning rate in CB1KOs is due solely to comparison with wildtype mice that are on average, more active.

CRs are not all-or-none; their amplitudes fluctuate from trial-to-trial and peak eyelid closure is typically timed to coincide with the presentation of the US. We analyzed the amplitude and timing of CRs in CB1KOs by analyzing trials in which the CS was presented alone, without a US. Like the percentage of CRs (*Figure 1F*), eyelid closures in CB1KOs (*Figure 1G*, red line) were smaller than the CRs elicited in the more active control animals (*Figure 1G*, dashed line), but comparable to their activity-matched controls (*Figure 1G*, solid lines). In addition, CRs of CB1KOs, like those of controls, were well-timed: they peaked at around the time that the US would have been presented (*Figure 1G*; see also *Figure 1—figure supplement 2*).

We previously showed for wildtype mice that both session-to session and trial-to-trial variation in the amplitude of CRs is positively correlated with locomotor speed (*Albergaria et al., 2018*). This was also true for global CB1KOs (trials: *Figure 1H*; sessions: *Figure 1—figure supplement 1H*). At a given walking speed, CR amplitudes were comparable for CB1KOs and their littermate controls (*Figure 1H*). However, there were differently skewed distributions of locomotor speeds across trials in the two genotypes (*Figure 1H* histogram). Therefore, to quantitatively compare CR amplitudes, we analyzed trials in which the animals were stationary, starting from onset of learning (*Figure 1I*). There was no significant difference in the amplitude of eyelid closure for stationary trials in CB1KO mice (*Figure 1I,J,K*; p=0.58).

Endocannabinoid signaling mediates short-term plasticity that acts on a timescale of tens of milliseconds up to several seconds (*Chevaleyre et al., 2006*; *Regehr et al., 2009*). We therefore investigated whether there might be more subtle effects on learned responses on these timescales in CB1KO mice (*Figure 1—figure supplement 2*). First, we investigated the timing of CRs to longer (500 ms) interstimulus intervals (*Perrett et al., 1993*; *Chettih et al., 2011*). Both CB1KOs and controls exhibited well-timed CRs on CS-only trials that peaked at around the time that the US would have been delivered (*Figure 1—figure supplement 2A*). There was no significant difference between genotypes in response timing for either the 300 ms (p=0.12) or 500 ms (p=0.24)

interstimulus intervals (*Figure 1—figure supplement 2B*). Next, we asked whether a lack of CB1 signaling could interfere with single-trial learning effects, in which response amplitudes are modulated based on the presence or absence of an air-puff US on the *previous* trial (*Figure 1—figure supplement 2C*, inset; *Medina and Lisberger, 2008*; *Yang and Lisberger, 2010*; *Khilkevich et al., 2016*; *Najafi and Medina, 2020*). Both CB1KOs and controls displayed larger amplitude CRs on trials following trials that included a puff (*Figure 1—figure supplement 2C,D*). Thus, even these more subtle features of delay eyeblink conditioning, both of which have been hypothesized to result from plasticity in the cerebellar cortex, were intact in CB1KO mice.

Taken together, the results so far suggest that the apparent impairments in eyeblink conditioning of CB1KO mice, both in terms of rate and amplitude of learning, can be fully accounted for by their hypoactivity. In other words, for CB1KO mice, behavioral state as measured by locomotor activity is a stronger determinant of learning than is genotype.

## Externally controlling locomotor activity rescues CB1R-related learning deficits

If impaired eyeblink conditioning in CB1KOs is solely a consequence of hypoactivity, then circumventing it by externally controlling locomotion (*Albergaria et al., 2018*) should be sufficient to rescue learning. Indeed, placing mice on a motorized treadmill that equalized locomotor activity for all animals and trials completely restored learning in CB1KOs (*Figure 2A,C*). In fact, there was now a trend toward faster learning in the knockouts, although this was not statistically significant. The average amplitude of eyelid responses was not statistically different between the two genotypes under these conditions (*Figure 2D,F*; p=0.82).

To bypass possible developmental or compensatory effects of CB1R deletion (*Berghuis et al., 2007*) and test the effects of acute suppression of CB1 receptor function, we investigated the effects of a CB1R antagonist on eyeblink conditioning on a motorized treadmill. Wildtype mice were injected intra-peritoneally with the CB1R antagonist AM251 before each training session. Similar to the locomotor phenotype of CB1KO mice, acute antagonist application reduced several measures of locomotor activity both across and within animals (*Figure 2—figure supplement 1*). However, there was no delay in learning in mice injected with the CB1R antagonist compared to vehicle-injected controls when tested on a motorized treadmill (*Figure 2B,C*). Rather, there was a trend toward faster learning in the presence of the antagonist (p=0.05). There was no difference in the amplitude of CRs (*Figure 2E–F*; p=0.32). In contrast, consistent with the strong influence of levels of locomotor activity on learning, on a self-paced treadmill learning was abolished not only following CB1R antagonist administration, but also in vehicle-injected animals, which were also severely hypoactive (*Figure 2—figure supplement 2*). This effect of systemic DMSO prevented us from assessing the effects of AM251 on learning on a self-paced treadmill. However, like the hypoactive CB1KOs, both vehicle and AM251-treated animals were able to walk successfully on the motorized treadmill, where we could assess the specific effects of AM251. Together, these results again point to locomotor activity, rather than CB1R function, as a critical determinant of learning.

## Parallel fiber CB1 receptors are not required for delay eyeblink conditioning

Impaired eyeblink conditioning in global CB1KOs has been hypothesized (*Kishimoto and Kano, 2006*; *Carey et al., 2011*) to be due to the loss of CB1Rs from parallel fibers, where they mediate several forms of synaptic plasticity, including parallel fiber LTD (*Kreitzer and Regehr, 2001a*; *Kreitzer and Regehr, 2001b*; *Safo and Regehr, 2005*; *Brenowitz and Regehr, 2005*; *Soler-Llavina and Sabatini, 2006*; *Carey et al., 2011*). In contrast, our results instead suggest that the apparent effects of CB1R deletion on eyeblink conditioning may arise as a secondary consequence of the hypoactivity of global CB1KOs. To directly test for a role of parallel fiber CB1Rs, we tested eyeblink conditioning in mice (G6KO) in which CB1Rs were selectively deleted from granule cells within the cerebellar cortex, whose axons form parallel fiber inputs to Purkinje cells and interneurons (*Figure 3A*; *Carey et al., 2011*; *Fünfschilling and Reichardt, 2002*; *Marsicano et al., 2003*).

Delay eyeblink conditioning was intact in granule-cell-specific CB1 knockout mice compared to littermate controls (cyan and blue lines, respectively; *Figure 3*). The rate of learning (*Figure 3C,E*), and the amplitude and timing of conditioned responses (*Figure 3D,F,G*; p=0.75), were all normal, both

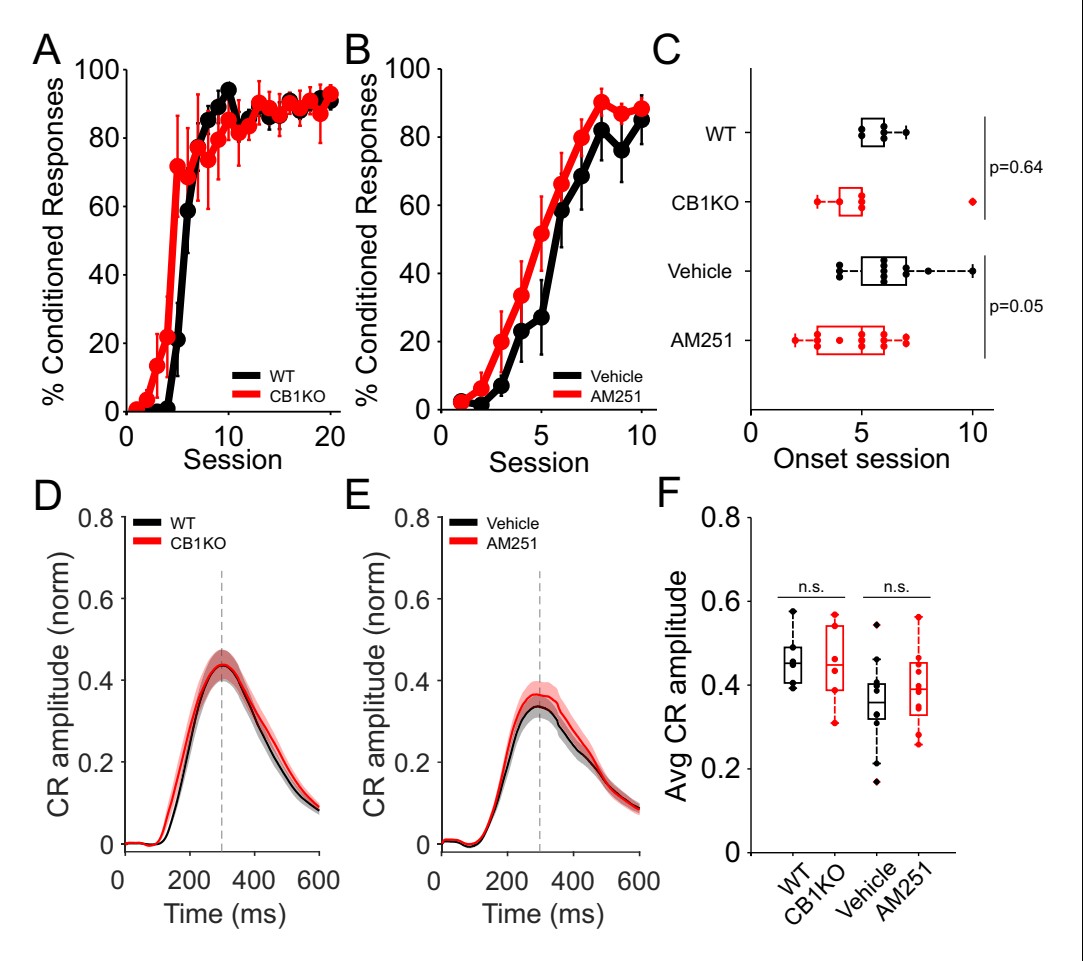

**Figure 2.** Equalizing locomotor activity with a motorized treadmill rescues CB1-related learning deficits.
(**A**) Average %CR learning curves of global CB1 knockout (red, N = 6) mice and littermate controls (black, N = 6), while running at a fixed speed (0.12 m/s) on a motorized treadmill. Error bars indicate SEM. (**B**) Average %CR learning curves of mice injected with vehicle solution (black, N = 12) or with the CB1R antagonist AM251 (red, N = 13), while running at a fixed speed (0.12 m/s). Error bars indicate SEM. (**C**) Average onset session of learning for animals in (**A**), wildtype (WT) and CB1KO mice (top two boxplots) and in (**B**), mice injected with vehicle or with AM251 (bottom two boxplots). There was no significant difference in learning onset session between WT and CB1KO (p=0.64). Average onset session between animals injected with vehicle or the antagonist AM251 was marginally different (p=0.05). Dots represent individual animals. Box indicates median and 25th to 75th percentiles, whiskers extend to the most extreme data points. (**D**) Average conditioned response (CR) eyelid traces of CS-only trials from all training sessions of control (black) and CB1 global knockout (red) animals. Shadows indicate SEM. Vertical dashed line represents the time that the US would have been expected on CS+US trials. (**E**) Average CR eyelid traces of CS-only trials from all training sessions of mice injected with vehicle (black) or AM251 (red). Shadows indicate SEM. Gray vertical line represents time of air-puff US. (**F**) Mean CR amplitudes from all training sessions of each experimental group: controls, CB1KO, animals injected with vehicle and animals injected with AM251 (from left to right). There was no significant difference in the average amplitude of WT and CB1KO mice (p=0.82, first two boxplots) or in mice injected with vehicle or AM251 (p=0.32, last two boxplots). Dots represent individual animals. Box indicates median and 25th to 75th percentiles, whiskers extend to the most extreme data points.

The online version of this article includes the following source data and figure supplement(s) for figure 2:

**Source data 1.** Source data for *Figure 2* and related supplements.
**Figure supplement 1.** CB1R antagonist AM251 causes hypoactivity, both across groups and within animals.
**Figure supplement 2.** Vehicle (DMSO) solution drives hypoactivity in mice, preventing them from learning on the self-paced treadmill.

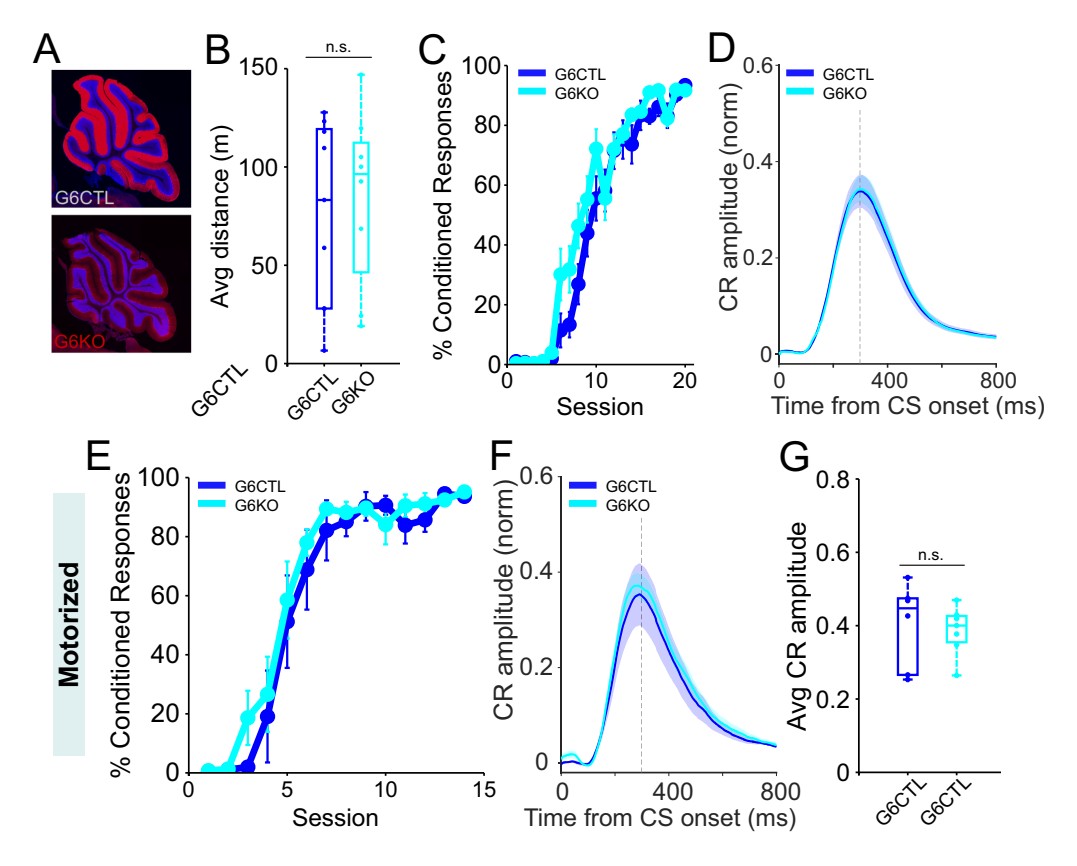

**Figure 3.** Eyeblink conditioning is intact in granule-cell-specific CB1 knockout mice. (A) Sagittal sections showing the selective elimination of CB1R (red) from cerebellar granule cells in G6KO and DAPI staining the granule-cell layer (blue). Images of the cerebellum of a representative control (upper panel) and granule-cell conditional CB1 knockout (bottom panel). (B) Average running distance of G6KO (cyan, N = 8) mice and their littermate controls (G6CTL, blue, N = 9) and on a self-paced treadmill across 20 learning sessions. The difference in average distance between groups was not significant (p=0.704). Box indicates median and 25th to 75th percentiles, whiskers extend to the most extreme data points. (C) Average %CR learning curves of granule-cell-specific CB1 knockout mice (G6KO, cyan, N = 8) and littermate controls (G6CTL, blue, N = 9), walking on the self-paced treadmill. Error bars indicate SEM. (D) Average conditioned response (CR) traces of CS-only trials from all training sessions on a self-paced treadmill for control (blue) and G6KO (cyan) animals. Shadows indicate SEM. Vertical dashed line represents the time that the US would have been expected on CS+US trials. (E) Average %CR learning curves of G6KO (cyan, N = 7) and G6CTL (blue, N = 6), while running at a fixed speed (0.12 m/s) on a motorized treadmill. Error bars indicate SEM. (F) Average CR eyelid traces of CS-only trials from all training sessions on a motorized treadmill of control (blue) and granule-cell-selective CB1 knockout (cyan) animals, on the motorized treadmill. Shadows indicate SEM. Vertical dashed line represents the time that the US would have been expected on CS+US trials. (G) Mean CR amplitudes from all training sessions. There was no significant difference in the average amplitude (p=0.75) of G6CTL (blue) vs. G6KO (cyan) animals. Dots represent individual animals. Box indicates median and 25th to 75th percentiles, whiskers extend to the most extreme data points. .

The online version of this article includes the following source data and figure supplement(s) for figure 3:

**Source data 1.** Source data for *Figure 3* and related supplements.

**Figure supplement 1.** Several measures of behavioral state reveal no differences between granule-cell-specific CB1 knockout mice and controls.

on the self-paced (*Figure 3C,D*) and motorized (*Figure 3E–G*) treadmills. Consistent with an extra-cerebellar source of altered behavioral state in global CB1KOs (*Dubreucq et al., 2013*), G6KO mice were not hypoactive (*Figure 3B*; p=0.7; *Figure 3—figure supplement 1*).

These results demonstrate that CB1Rs on cerebellar parallel fibers are dispensable for delay eyeblink conditioning.

## CB1Rs are dispensable for locomotor learning

There are many similarities between learning mechanisms for eyeblink conditioning and other forms of cerebellum-dependent learning such as motor adaptation (*Raymond et al., 1996*; *Raymond and*

*Medina, 2018*; *Gao et al., 2012*). However, there are also important differences, particularly in the time course of learning, which could be particularly relevant in the context of CB1-mediated plasticity, which typically acts on much shorter time scales than eyeblink conditioning, which takes days. We therefore investigated whether CB1Rs might play a role in a rapid form of cerebellum-dependent learning, locomotor adaptation on a split-belt treadmill (*Figure 4*; *Yanagihara and Kondo, 1996*; *Morton and Bastian, 2006*; *Darmohray et al., 2019*).

Motor coordination is generally thought to be largely spared in CB1KOs (*Zimmer et al., 1999*; *Steiner et al., 1999*; *Bilkei-Gorzo et al., 2005*; *Kishimoto and Kano, 2006*; *Varvel and Lichtman, 2002*). They are not overtly ataxic (*Video 1*) and show normal rotarod performance at the ages tested here (*Zimmer et al., 1999*; *Bilkei-Gorzo et al., 2005*; *Kishimoto and Kano, 2006*). However, since global CB1KOs are hypoactive, and since split-belt treadmill adaptation involves learned changes in interlimb coordination, we first performed a detailed analysis of locomotor kinematics during overground walking (*Machado et al., 2015*; *Machado et al., 2020*) to detect any potential subtle locomotor impairments in CB1KOs (*Video 1*). Most gait parameters were normal in CB1KOs during overground walking (*Figure 4—figure supplement 1A–C*), although there were some subtle differences, such as paws not lifting as high (*Figure 4—figure supplement 1B*, right) and a small tendency toward a walking gait pattern rather than a trot at lower speeds (*Figure 4—figure supplement 1C*, middle). Notably, none of these locomotor differences were observed in G6KO mice (*Figure 4—figure supplement 1D–F*, *Figure 4—figure supplement 2*). Further, these differences do not match the pattern of deficits found in mouse models of cerebellar ataxia (*Figure 4—figure supplement 2*; *Lalonde and Strazielle, 2007*; *Machado et al., 2015*; *Vinueza Veloz et al., 2015*; *Machado et al., 2020*).

In locomotor adaptation on a split-belt treadmill, animals learn to regain overall gait symmetry in response to a perturbation that imposes unequal speeds on the two sides of the body (*Figure 4A*). Learning is specific to measures of interlimb coordination, which compare the symmetry of gait parameters that depend on multiple limbs (*Reisman et al., 2005*; *Darmohray et al., 2019*). Individual limb parameters, such as stride length (how far forward one paw moves during a swing phase; *Figure 4B–D*), change abruptly upon acute changes in belt speed, but do not show learning: they remain constant throughout the exposure to split-belt walking and return to normal once symmetrical belt speeds are restored. In contrast, interlimb coordination measures, such as step length (the relative anterior-posterior distance at stance onset between a paw and its contralateral counterpart; *Figure 4E–G*, see Methods), show gradual improvements in symmetry throughout the split-belt period, and prominent aftereffects in the opposite direction upon its cessation (*Reisman et al., 2005*; *Darmohray et al., 2019*). These learned changes in interlimb coordination, but not the acute resetting of individual limb parameters upon splitting of the belts, are cerebellum-dependent (*Morton and Bastian, 2006*; *Darmohray et al., 2019*).

Like controls, both global and granule-cell-specific CB1KOs displayed acute individual (*Figure 4C,D*) and interlimb (*Figure 4F,G*) asymmetries upon the splitting of the belts. This initial response differed in global, but not G6KO mice: individual limb stride lengths became more asymmetric (*Figure 4C*; $t_{(48)}$ = 3.64, p=0.007), while initial interlimb asymmetries were smaller than expected (step length, *Figure 4F,H*; $t_{(48)}$ = 2.63, p=0.01). Inspection of locomotor kinematics during baseline trials, when the belts were moving at the same speed, reveals subtle differences in treadmill walking that could underlie these different reactions to the splitting of the belts (*Video 2*; *Figure 4—figure supplement 3A–C*). None of these differences were observed in granule-cell-specific CB1KOs (*Figure 4D,G*; *Figure 4—figure supplement 3D–F*; stride length: $t_{(48)}$ = 0.20 *p*=0.84; step length: $t_{(48)}$ = −1.03, p=0.30).

Despite the variation in the acute response to the splitting of the belts, both global and granule-cell-specific CB1KOs learned to walk more symmetrically over the course of exposure to split-belt walking (*Figure 4F,G*, gray shaded areas; *Figure 4H–K*). Further, upon resumption of tied-belt walking, both global and granule-cell-specific CB1KOs exhibited step-length aftereffects in the form of asymmetries in the opposite direction (*Figure 4F–K*). Locomotor learning in global CB1KOs, measured as both the improvements in symmetry over the course of split-belt walking and the step-length aftereffects, was comparable to that of controls, in terms of both absolute magnitude (*Figure 4H*; late − early: $t_{(48)}$ = 0.7, p=0.45; aftereffect $t_{(48)}$ = −1.06, p=0.29), and in the percent of the initial error that was compensated for (*Figure 4I,K*; late − early: $t_{(16)}$ = −1.3, p=0.2; aftereffect: $t_{(16)}$ = 0.6, p=0.57). All measures were normal in G6KOs (abs. magnitude: late − early: $t_{(48)}$ = 0.68,

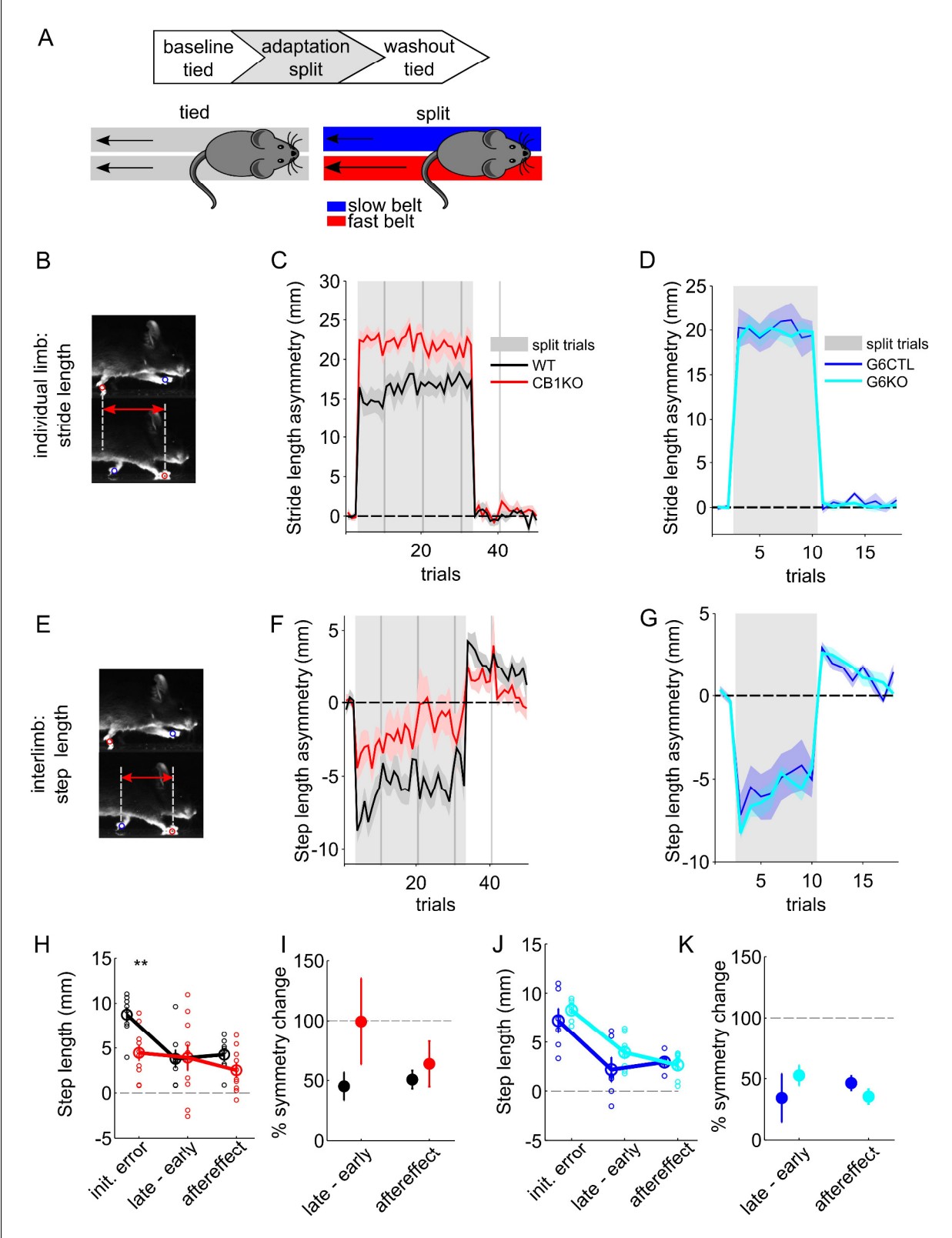

**Figure 4.** Locomotor learning in global and granule-cell-specific CB1 knockout mice. (**A**) Experimental protocol for split-belt locomotor adaptation. Mice underwent adaptation protocols consisting of baseline, split-belt (adaptation) and washout phases. Belt speeds were equal ('tied') in baseline trials and were then abruptly split (2:1 speed ratio) for the adaptation phase, before returning to the original, symmetrical tied-belt speed in the washout phase. (**B**) The intra-limb parameter 'stride length' is the total forward motion of an individual limb from lift-off (swing) to touch-down (stance).
*Figure 4 continued on next page*

*Figure 4 continued*

(C) Average front limb stride length asymmetry (fast-slow) over trials for global CB1 knockout mice (CB1 KO, N = 10, red) and littermate controls (CB1 WT, N = 8, black). Gray shaded patch indicates split-belt trials. Vertical gray bars indicate session breaks. (D) Same as (C) for granule-cell-specific CB1 knockout mice (G6KO, N = 8, cyan) and their littermate controls (G6CTL, N = 6, blue). (E) The interlimb parameter 'step length' is how far forward one paw is relative to its contralateral pair at stance onset. (F) Average front limb step-length asymmetry (fast-slow) over trials for global CB1 knockout mice and littermate controls, plotted as in (B). (G) Same as (F) for granule-cell-specific CB1 knockout mice and their littermate controls. (H) Average step-length asymmetries at three key experimental phases: initial error during the first split-belt trial, the change in step-length asymmetry over the split period (defined as the difference in step-length asymmetries in the last minus the first split-belt trial), and aftereffect (first trial upon returned to tied-belt condition), +/- SEM, for global CB1KOs (red) and littermate controls (black). The signs of the changes over split and aftereffects have been inverted in order to allow direct comparison of magnitudes across the three epochs. Individual animals are shown with smaller, open circles. The change over the split-belt period and the aftereffects, which both reflect learned changes in step symmetry, are comparable in the two genotypes. (I) Average changes in step symmetry (+/- SEM) are plotted as a percent of the average initial error, for global CB1KO mice and littermate controls. Line at 100% shows complete adaptation. There is no deficit in learning in the CB1KOs. (J) Same as (H) for granule-cell-specific CB1 knockouts (cyan) and littermate controls (blue). Individual animals are shown with smaller, open circles. (K) Same as (H) but for granule-cell-specific CB1 knockouts and littermate controls.

The online version of this article includes the following source data and figure supplement(s) for figure 4:

**Source data 1.** Source data for *Figure 4* and related supplements.
**Figure supplement 1.** Analysis of overground locomotion in global and granule-cell-specific CB1KO mice.
**Figure supplement 2.** Linear discriminant analysis separates CB1KO mice from two mutants with cerebellar ataxia.
**Figure supplement 3.** Analysis of treadmill locomotion in global and granule-cell-specific CB1KO mice.

p=0.49; aftereffect $t_{(48)}$ = 0.05, p=0.95; % initial error:: late – early: $t_{(12)}$ = 1.27, p=0.22; aftereffect $t_{(12)}$ = −0.94, p=0.36).

Thus, despite some subtle differences in locomotor kinematics that were present in global, but not granule-cell-specific CB1KO mice, cerebellum-dependent locomotor learning was intact in both global and G6KOs.

## Discussion

The well-described cerebellar circuit architecture places strong constraints on the sites and mechanisms for learning. Despite this apparent simplicity, the contributions of various plasticity mechanisms, including LTD between parallel fibers and Purkinje cells, to cerebellar learning remain controversial (*Carey, 2011*; *Schonewille et al., 2011*; *Johansson et al., 2018*). Results from gene knockout experiments have contributed to these controversies (*Aiba et al., 1994*; *De Zeeuw et al., 1998*; *Koekkoek et al., 2003*; *Schonewille et al., 2011*). Several previous studies have demonstrated that cannabinoids, strong modulators of synaptic transmission that are required for several forms of plasticity, influence cerebellum-dependent associative learning (*Kishimoto and Kano, 2006*; *Steinmetz and Freeman, 2010*; *Steinmetz and Freeman, 2020*). Here we show that this influence is exerted indirectly, through alteration of behavioral state; CB1 receptors themselves are dispensable for two distinct forms of cerebellar learning.

Multiple lines of evidence presented here support the conclusion that the apparent effects of CB1 deletion on cerebellar learning arise as a secondary consequence of the hypoactivity of global CB1KOs, rather than through direct effects on cerebellar plasticity. First, eyeblink conditioning deficits in CB1KO mice were fully accounted for by differences in locomotor activity, across both animals and trials (*Figure 1*). Second, eyeblink conditioning in global CB1KOs was fully rescued by equalizing locomotor activity with a motorized treadmill (*Figure 2*). Third, a systemic CB1-antagonist did not impair eyeblink conditioning on a motorized treadmill (*Figure 2*). Fourth, CB1KOs were able to learn to walk more symmetrically in a cerebellum-dependent locomotor learning task (*Figure 4*). Finally, mice lacking CB1 receptors in cerebellar granule cells

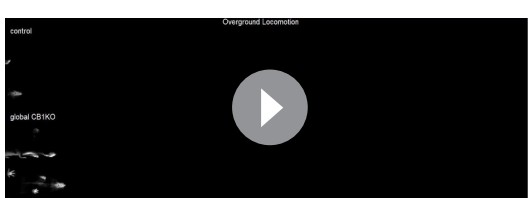

**Video 1.** Example movie of littermate control mouse (top) and global CB1KO mouse (bottom) walking on LocoMouse overground setup.
https://elifesciences.org/articles/61821#video1

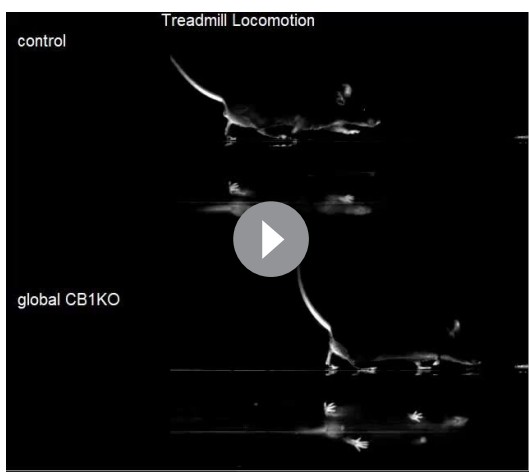

**Video 2.** Example movie of littermate control mouse (top) and global CB1KO mouse walking on LocoMouse treadmill during 'tied' belt condition (0.275 m/s). https://elifesciences.org/articles/61821#video2

exhibited none of the behavioral phenotypes observed in global knockouts (*Figures 3* and *4*).

Knockout animals have been used extensively to investigate the role of specific receptors and signaling pathways in complex behaviors such as learning and memory (*Crawley, 2008*; *Dubnau and Tully, 1998*; *Picciotto and Wickman, 1998*). The sometimes-conflicting findings from this approach are often attributed to its widely-acknowledged drawbacks, including limited cellular and temporal specificity, and the possibility of compensatory mechanisms (*El-Brolosy and Stainier, 2017*). In this case, we present a different confound: a receptor that appears to be required for learning in one behavioral context, is not required in another. Altering behavioral state – whether via the cannabinoid system, or by exposure to a motorized treadmill – profoundly modulates the capacity for learning. As a consequence, externally controlling locomotor activity rescues the effect of CB1 receptor deletion, revealing a complex interplay between genes, behavior, and learning (*Eisener-Dorman et al., 2009*; *El-Brolosy and Stainier, 2017*; *Picciotto and Wickman, 1998*; *Crawley, 2008*).

Surprisingly, when we corrected for differences in locomotor activity, we often observed a small but consistent enhancement of learning in CB1KOs. This was true on the self-paced treadmill (*Figure 1E,F*) as well as on the motorized treadmill (*Figure 2A,C*). This could be due to compensatory mechanisms upregulated in response to parallel fiber CB1 deletion, although interestingly, a similar trend was also observed following acute treatment with a CB1R antagonist (*Figure 2B,C*). An alternative possibility is that this small enhancement is a consequence of decreased arousal (*Figure 1—figure supplement 1D*; *Albergaria et al., 2018*). Although locomotor activity levels generally increase with heightened arousal, as measured by pupil size (*Vinck et al., 2015*; *Mineault et al., 2016*; *Reimer et al., 2014*; *Reimer et al., 2016*), by controlling for differences in locomotion we previously demonstrated that increased arousal on its own has a small negative influence on learning (*Albergaria et al., 2018*). Thus the subtle enhancement of learning and the smaller pupil sizes in CB1KO mice (*Figure 1—figure supplement 1*) are consistent with the possibility that the altered behavioral state of the CB1KOs could have two competing effects on learning: one detrimental, due to decreased locomotor activity, and one beneficial, due to decreased arousal. Under physiological conditions, however, the enhancement by locomotor activity is a much stronger determinant of learning (*Albergaria et al., 2018*).

The modulation of behavioral state in CB1KO mice is likely to be of extra-cerebellar origin. Deletion of CB1 receptors from interneurons in the ventral tegmental area was identified as the source of hypoactivity in CB1KO mice (*Dubreucq et al., 2013*). CB1KOs also exhibit a number of other behavioral differences, including altered feeding and anxiety levels (*Zimmer et al., 1999*; *Degroot and Nomikos, 2004*; *Ravinet Trillou et al., 2004*). It is possible that CB1Rs on non-granule-cell types within the cerebellum, such as molecular layer interneurons or climbing fibers (*Kreitzer and Regehr, 2001b*; *Diana and Marty, 2004*) may contribute either to hypoactivity and/or to the subtle motor phenotype we observed in mice walking on a motorized treadmill (*Video 2*, *Figure 4—figure supplement 3*). However, regardless of the origin of the hypoactivity, correcting for it reveals that CB1Rs are not required for eyeblink conditioning.

The finding that all behavioral measures were normal in G6KOs suggests that all forms of parallel fiber-mediated CB1-dependent plasticity at synapses onto either Purkinje cells or interneurons (*Kreitzer and Regehr, 2001a*; *Brown et al., 2003*; *Brenowitz and Regehr, 2005*; *Safo and Regehr, 2005*; *Soler-Llavina and Sabatini, 2006*; *Carey et al., 2011*) are dispensable for eyeblink conditioning as well as cerebellum-dependent locomotor coordination and learning. This seems unlikely to be

solely due to compensation in the knockouts, since a CB1 antagonist, which induces hypoactivity, similarly caused no impairment in eyeblink conditioning on a motorized treadmill.

In pointing toward an extracerebellar role for CB1 receptors in modulating cerebellar learning, our results are consistent with previous findings that systemic CB1 antagonists impair eyeblink conditioning (*Kishimoto and Kano, 2006*; *Steinmetz and Freeman, 2010*), but localized infusion in the cerebellar cortex does not (*Steinmetz and Freeman, 2020*). While both systemic (*Steinmetz and Freeman, 2016*) and intracerebellar (*Steinmetz and Freeman, 2020*) CB1 agonists impair acquisition, this is likely to be due to acute suppression of neurotransmitter release, rather than reflecting specific effects on learning. Indeed, our results suggest that endocannabinoid signaling within the cerebellar cortex may be less important for learning-related plasticity than for regulating overall levels of excitability and transmission (*Marsicano et al., 2003*; *Regehr et al., 2009*).

Since previous studies have demonstrated the necessity of CB1 receptors in parallel fiber LTD (*Safo and Regehr, 2005*; *Carey et al., 2011*), our results would seem to argue against a critical role for parallel fiber LTD in cerebellar learning. While we cannot rule out the possibility that a CB1-independent form of LTD could be invoked in vivo, there are a host of other plasticity mechanisms within the cerebellar circuit that could provide potential substrates for learning (*Hansel et al., 2001*; *Carey, 2011*; *Gao et al., 2012*; *Johansson et al., 2015*). However, we emphasize that a broader point made by our results is that such reductionist approaches to the complex system that is the brain of a behaving animal should be taken with caution. Myriad plasticity mechanisms can be induced in vitro, and variations in induction protocols and other factors may alter the specific cellular pathways involved under various conditions. Neuromodulators like cannabinoids have multiple effects on synaptic transmission, plasticity, and behavioral state, and as we underscore here, untangling these effectively requires more than one approach.

In conclusion, CB1 receptors modulate cerebellum-dependent associative learning via indirect effects on behavioral state, and not via CB1-mediated parallel fiber plasticity. A large body of recent work has demonstrated profound effects of behavioral state, including locomotor activity and arousal, on brain function (*Niell and Stryker, 2010*; *Bennett et al., 2013*; *Ayaz et al., 2013*; *Polack et al., 2013*; *Erisken et al., 2014*; *Vinck et al., 2015*; *Pinto et al., 2013*; *Hardcastle et al., 2017*; *Musall et al., 2019*; *Stringer et al., 2019*) and capacity for learning (*Albergaria et al., 2018*). Our findings extend these previous studies to establish alterations in behavioral state as a powerful, independent way that individual genes can contribute to complex behaviors such as learning and memory.

## Materials and methods

### Animals

All procedures were carried out in accordance with the European Union Directive 86/609/EEC and approved by the Champalimaud Centre for the Unknown Ethics Committee and the Portuguese Direcção Geral de Veterinária (Ref. No. 0421/000/000/2015). Mice were kept on a reversed 12 hr light/12 hr dark cycle, in standard cages with typically 2–4 animals per cage. They had access to food and water ad libitum. All procedures were performed in male and female mice approximately 10–14 weeks of age.

### Global and conditional knockouts

Global CB1R knockout mice (*Cnr1 -/-*, here termed CB1KO) (*Zimmer et al., 1999*) and their littermate controls (*Cnr1 +/+*) were obtained by crossing heterozygous breeding pairs. *Gabra6-Cre;Cnr1 flox/flox* mice were generated by crossing mice (*Gabra6-Cre*) in which Cre recombinase expression was driven by the promoter of the alpha6 subunit of the GABA$_A$ receptor and was specific to granule cells within the cerebellar cortex (*Fünfschilling and Reichardt, 2002*), with mice (*Cnr1 flox/flox*, *Marsicano et al., 2003*) carrying floxed alleles of the *Cnr1* gene that encodes the CB1R receptor. *Gabra6-Cre;Cnr1 flox/flox* mice have been previously characterized and shown to lack parallel fiber LTD, as well as CB1R-mediated short-term forms of parallel-fiber-Purkinje cell plasticity (*Carey et al., 2011*). All lines were kept in a C57BL/6 J background.

## Surgical procedures

In all our surgeries, animals were anesthetized with isoflurane (4% induction and 0.5–1% for maintenance), placed in a stereotaxic frame (David Kopf Instruments, Tujunga, CA) and a custom-cut metal head plate was glued to the skull with dental cement (Super Bond – C and B). After any surgical procedure, mice were monitored and allowed ~1–2 days of recovery.

## Drugs

Mice were injected intra-peritoneally with AM251 (A6226, Sigma; 3 mg/kg; *Corbillé et al., 2007*; *Sousa et al., 2011*; *Xi et al., 2008*) or vehicle, 30 min before each training session. AM251 was suspended in saline with 5% DMSO and 1% Tween 80, at a concentration of 3 mg/ml. Control mice were injected with vehicle consisting of saline with 5% DMSO and 1% Tween 80. The vehicle alone reduced locomotor activity levels but did not impair ability of the mice to walk on the motorized treadmill (*Figure 2—figure supplement 2*), and AM251 induced additional hypoactivity beyond the effects of vehicle (*Figure 2—figure supplement 1*).

## Histology

To confirm CB1R expression in the different mouse lines, animals were perfused transcardially with 4% paraformaldehyde and their brains removed. Sagittal sections (50 µm thick) were cut in a Cryostat and stained with a polyclonal guinea pig antibody raised against the last 31 amino acids of the CB1R C-terminal (from Frontier Institute co., ltd) and DAPI. Sections were mounted on glass slides with Vectashield mounting medium and imaged with a 10x objective.

## Eyeblink conditioning

The experimental setup was based on previous work (*Albergaria et al., 2018*; *Chettih et al., 2011*). For all behavioral experiments, mice were head-fixed but could walk freely on a Fast-Trac Activity Wheel (Bio-Serv) and habituated to the behavioral setup for at least 4 days prior to training. To externally control the speed of the treadmill, a DC motor with an encoder (Maxon) was used. For experiments on the motorized treadmill, mice were additionally habituated to walk at the target speed until they walked normally and displayed no external signs of distress. There was no difference across genotypes in the amount of habituation time.

Locomotor activity was measured using an infra-red reflective sensor placed underneath the treadmill. Eyelid movements of the right eye were recorded using a high-speed monochromatic camera (Genie HM640, Dalsa) to monitor a 172 × 160 pixel region, at 900fps. Custom-written software using LabVIEW, together with a NI PCIE-8235 frame grabber and a NI-DAQmx board (National Instruments), was used to trigger and control all the hardware in a synchronized manner.

Acquisition sessions consisted of the presentation of 100 CS-US paired trials and 10 CS-only trials, which allow for the analysis of the kinematics of CRs without the masking effect that comes from the US-elicited reflex blink. The 110 trials were separated by a randomized inter-trial interval (ITI) of 5–20 s. In each trial, CS and US onsets were separated by a fixed interval (ISI) of 300 ms and both stimuli co-terminated. The experiments in *Figure 1—figure supplement 2* were conducted in dedicated 'test sessions' following acquisition, in which only 50% of the trials contained an air-puff US, in order to assess the influence of US presence/absence on single-trial learning.

For all training experiments, the unconditioned stimulus (US) was an air-puff (40 psi, 50 ms) controlled by a Picospritzer (Parker) and delivered via a 27G needle positioned ~0.5 cm away from the cornea of the right eye of the mouse. The direction of the air-puff was adjusted for each session of each mouse so that the unconditioned stimulus elicited a normal eye blink. The CS had a 350 ms duration and was a white light LED (5 mW) positioned ~3 cm directly in front of the mouse.

The video from each trial was analyzed offline with custom-written software using MATLAB (MathWorks). The distance between eyelids was calculated frame by frame by thresholding the grayscale image and extracting the minor axis of the ellipse that delineated the eye. Eyelid traces were normalized for each session, ranging from 1 (full blink) to 0 (eye fully open). Trials were classified as CRs if the eyelid closure reached at least 0.1 (in normalized pixel values) and occurred between 100 ms after the time of CS onset and the onset of US.

## Analysis of locomotor coordination

Locomotor coordination was assessed using our previously described LocoMouse setup, a tracking and analysis system for freely moving mice (*Machado et al., 2015*). Briefly, mice walked across a glass corridor, with a mirror placed at 45 deg below the mouse, so that a single high-speed camera recorded both bottom and side views. Individual trials consisted of single crossings of the corridor. Mice initiated trials by walking back and forth between two dark 'home' boxes on each end of the corridor. Tracking and gait analysis was performed offline.

Tracking data was first broken down into strides using a simple peak detection algorithm (*Machado et al., 2015*). For the analyses shown in *Figure 4—figure supplement 2*, we computed a range of locomotor parameters to compare global CB1KO mice to *Purkinje cell degeneration* (*pcd*) and *reeler* mutants with cerebellar atrophy and consequent ataxia (*Lalonde and Strazielle, 2007*; *Mullen et al., 1976*; *Falconer, 1951*; *Machado et al., 2015*; *Machado et al., 2020*). We included 45 gait parameters describing the movement of individual limbs, interlimb coordination, body trajectories and variability. The reported gait parameters have been previously described (*Machado et al., 2015*; *Machado et al., 2020*). Gait parameter descriptions were computed as follows:

### Individual limb parameters

Individual limb parameters for both front right (FR) and hind right (HR) paws were included in all analyses.
Stride duration: time between two consecutive stance onsets.
Cadence: inverse of stride duration.
Swing velocity: x displacement of single limb during swing phase divided by the swing duration.
Stride length: x displacement from touch-down to touch-down of single limb.
Duty factor: stance duration divided by stride duration.
Trajectories: (x,y,z) trajectories were aligned to swing onset and resampled to 100 equidistant points using linear interpolation. Interpolated trajectories were then binned by speed and the average trajectory was computed for each individual animal and smoothed with a Savitzky-Golay first-order filter with a 3-point window size.
Instantaneous swing velocity: the derivative of swing trajectory.
Variability: All variability analyses were based on coefficients of variation (CV).

### Interlimb and whole-body coordination parameters

Base of support: width between the two front and two hind paws during stance phase.
Stance phase: relative timing of limb touchdowns to stride cycle of reference paw (FR). Calculated as: stance time - stance time$_{reference\ paw}$/stride duration. We report stance phase as left-right (LR) and front-hind (FH) phase.
Supports: Support types were categorized by how many and which paws were on the ground, expressed as a percentage of the total stride duration for each stride. Paw support categories include 3-paw, 2-paw diagonal, 2-paw other/non diagonal (homolateral and homologous).
Double support: for each limb is defined as the percentage of the stride cycle between the touch-down of a reference paw to lift-off of the contralateral paw.
Center of oscillation: midpoint between swing and stance x positions relative to body center.
Step length: displacement of one limb relative to its contralateral homolog at stance onset.

For linear discriminant analysis (LDA, *Figure 4—figure supplement 2A*), each observation was data from one mouse locomoting at a specific speed and features are z-scored gait parameters. Since, LDA assumes independence within the feature space, PCA was applied first to address inter-variable correlation and avoid overfitting. PCA was performed by eigenvalue decomposition of the data covariance matrix. The first 10 PCs explained 86% of the variance and the data projected onto these 10 PCs was used as input to the LDA. The end contributions of the initial gait parameters to the two LD axes were obtained by multiplying the PCA mapping by the LDA mapping.

## Locomotor learning on a split-belt treadmill

Split-belt locomotor adaptation experiments and analyses were performed on a modified version of the LocoMouse setup (*Machado et al., 2015*) as previously described (*Darmohray et al., 2019*).

Two motor-driven transparent treadmill belts independently imposed the walking speed on the two sides of the body. Split-belt locomotor adaptation experiments consisted of 'baseline' tied, split-belt, and 'washout' tied-belt trials. All trials were one minute in duration, with brief periods in which the motors were off, in between trials.

Granule-cell-specific CB1KO mice and their littermate controls were run in a single session adaptation protocol (two tied trials, eight split trials; eight tied trials). For these mice, split-belt trial speeds were at a 2.14:1 ratio: 0.175 m/s (slow) and 0.375 m/s (fast). Global CB1KO mice and their littermate controls underwent a longer, multi-session adaptation protocol consisting of 10 trials per day (session 1: three tied, seven split; sessions 2–3: 10 split; session 4: three split, seven tied; session 5: 10 tied), with lower overall belt speeds (tied: 0.2 m/s; slow: 0.125 m/s fast: 0.275 m/s).

For split-belt locomotor adaptation analyses (*Figure 4*), we compared mutants and littermate controls on individual and interlimb coordination parameters. For individual limb analyses, we compared the two groups on their initial response to split-belt walking by assessing how stride length and step-length symmetry scaled on the first split-belt trials. Limb speeds during stance were monitored throughout the split-belt period to ensure that animals were consistently maintaining split-belt walking; two animals were excluded from the global CB1KO experiment for not meeting this criterion. To assess learned changes in interlimb coordination, we compared the change in symmetry from early to late split (late – early) and aftereffects (first washout trial – average baseline) of mutants and littermate controls. For percent symmetry change plots, changeover split trials (late – early) and aftereffects were normalized by the average (group) initial error.

## Statistics

Statistical analyses were performed using the Statistics toolbox in MATLAB and R. For the correlation between speed and CR amplitude (*Figure 1H*), we used a mixed model approach. We specified random slopes and intercepts models and included mouse/subject as a random covariate using the lme2 package (*Bates, 2005*). We report F tests (ANOVA) with Satterthwaite degrees of freedom correction; reported post-hoc analyses are t-tests with Tukey corrections for multiple comparisons. For the correlation between average distance and onset session (*Figure 1E*), as well as session-to-session changes in distance and CR amplitude (*Figure 1—figure supplement 1H*), we used linear regression. To compare the average distance between animals of each genotype (*Figure 1D*; *Figure 3B*), the differences between average CR peak amplitudes (*Figure 1—figure supplement 2B*), onset learning sessions (*Figure 2C*) and the average amplitudes of eyelid closure (*Figure 1K*; *Figure 2F*; *Figure 3G*), we used Student's unpaired t-tests. For comparing behavioral state parameters between CB1KOs and littermate controls (*Figure 1—figure supplement 1B–E*), as well as between animals injected with AM251 or vehicle solution (*Figure 2—figure supplement 1A–D*) and granule-cell-specific CB1KOs and respective controls (*Figure 3—figure supplement 1A–D*), we used the same test. For within animal comparisons (*Figure 2—figure supplement 1E-G*, *Figure 1—figure supplement 2D*), we performed a Student's paired t-test.

For locomotor adaptation analyses, we used mixed models to test for fixed effects of genotype (CB1KOs and littermate controls) and experimental phase on the asymmetry of each gait parameter (either stride length, step length). Experimental phase had three levels including early split (initial error), change over split trials (late - early) and aftereffects. Early and late split were the first and last split-belt trials. Aftereffects were the first post-split tied-belt trial. We specified random slopes and intercepts models and included mouse/subject as a random covariate using the *lme4* package (*Bates, 2005*). Reported statistics are post-hoc t-tests with Tukey corrections for multiple comparisons and were conducted using the *lsmeans* package in R (*Lenth, 2016*). All statistical comparisons are conducted on animal averages (i.e., each animal has one observation per level(s) of the independent variable).

All t-tests were two-tailed. Differences were considered significant at *p<0.05, **p<0.01, and ***p<0.001. No sample size calculation was performed, Although sample sizes were not pre-determined with sample size calculations, they were based on related previous research (*Heiney et al., 2014*; *Albergaria et al., 2018*; *Darmohray et al., 2019*), and statistical tests were run after data collection. Mice were assigned to specific experimental groups according to their genotype and without bias.

## Acknowledgements

We thank Tracy Pritchett, Ana Vaz, and the Champalimaud Histology Platform for technical support. Sofia Araújo, Catarina Almeida, Sofia Cohen, and Teresa Duarte assisted with some experiments. We thank Wade Regehr, Javier Medina, and Rui Costa for support during the initial planning of this project. We are grateful to the members of the Carey lab and the Champalimaud Neuroscience Program for helpful discussion.

This work was supported by Howard Hughes Medical Institute International Early Career Scientist Grant #55007413 (to MRC), European Research Council Starting Grant #640093 (to MRC), Congento LISBOA-01–0145-FEDER-022170, co-financed by FCT (Portugal) and Lisboa2020 under the PORTU-GAL2020 agreement, and fellowships from the Portuguese Fundação para a Ciência e a Tecnologia SFRH/BD/77686/2011 (to CA), SFRH/BD/105949/2014 (to NTS), SFRH/BD/86265/2012 (to DD).

## Additional information

### Competing interests

Megan R Carey: Reviewing editor, *eLife*. The other authors declare that no competing interests exist.

### Funding

| Funder | Grant reference number | Author |
| --- | --- | --- |
| Howard Hughes Medical Institute | International Early Career Scientist 55007413 | Megan R Carey |
| European Commission | European Research Council Starting Grant 640093 | Megan R Carey |
| Fundação para a Ciência e a Tecnologia | SFRH/BD/77686/2011 | Catarina Albergaria |
| Fundação para a Ciência e a Tecnologia | SFRH/BD/105949/2014 | N Tatiana Silva |
| Fundação para a Ciência e a Tecnologia | SFRH/BD/86265/2012 | Dana M Darmohray |

The funders had no role in study design, data collection and interpretation, or the decision to submit the work for publication.

### Author contributions

Catarina Albergaria, Conceptualization, Formal analysis, Investigation, Visualization, Writing - original draft, Writing - review and editing; N Tatiana Silva, Dana M Darmohray, Formal analysis, Investigation, Writing - original draft, Writing - review and editing; Megan R Carey, Conceptualization, Supervision, Funding acquisition, Visualization, Writing - original draft, Project administration, Writing - review and editing

### Author ORCIDs

Catarina Albergaria (iD) https://orcid.org/0000-0001-8257-3600
Megan R Carey (iD) https://orcid.org/0000-0002-4499-1657

### Ethics

Animal experimentation: All procedures were carried out in accordance with the European Union Directive 86/609/EEC and approved by the Champalimaud Centre for the Unknown Ethics Committee and the Portuguese Direcção Geral de Veterinária (Ref. No. 0421/000/000/2015).

### Decision letter and Author response

Decision letter https://doi.org/10.7554/eLife.61821.sa1
Author response https://doi.org/10.7554/eLife.61821.sa2

## Additional files

**Supplementary files**
• Transparent reporting form

### Data availability

All data generated or analysed during this study are included in the manuscript and supporting files. Source data files have been provided for all figures and supplements.

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
