## [Decision Letter]

**Acceptance summary:**

CB1 signaling is known to influence short- and long-term plasticity at the parallel fiber-Purkinje cell synapses, and CB1 antagonists also influence eyeblink conditioning, so the assumption has been that these two observations were causally related. Here, the authors upend this view, providing convergent lines of evidence that the effects of a CB1 antagonist on eyeblink conditioning are instead indirectly mediated by its effects on locomotion, which this lab has previously shown to influence eyeblink conditioning. There are significant implications for understanding the role of plasticity at the parallel fiber-Purkinje cell synapses in cerebellum-dependent learning. More generally, this work highlights the importance of considering global effects such as behavioral state when analyzing specific behaviors or learning and memory tasks and, as such, will be of interest to those beyond the endocannabinoid or cerebellar subfields.

**Decision letter after peer review:**

Thank you for submitting your article "Altered behavioral state accounts for the modulation of cerebellar learning by cannabinoid receptors" for consideration by *eLife*. Your article has been reviewed by three peer reviewers, including Jennifer L Raymond as the Reviewing Editor and Reviewer #1, and the evaluation has been overseen by Kate Wassum as the Senior Editor. The following individuals involved in review of your submission have agreed to reveal their identity: John Freeman (Reviewer #2); Yan Yang (Reviewer #3).

The reviewers have discussed the reviews with one another and the Reviewing Editor has drafted this decision to help you prepare a revised submission.

Summary:

Signaling through endocannabinoid (CB1) receptors has been shown to influence several forms of synaptic plasticity in the cerebellum and also to influence cerebellum-dependent learning tasks, including eyeblink conditioning. Here, the authors show quite convincingly that the effects of manipulating CB1 signaling on eyeblink conditioning are mediated, not directly through effects on cerebellar plasticity, as one would assume, but indirectly through the effects of CB1 signaling on the behavioral state of the animal, and, more specifically, the level of locomotor activity. The study design is strong, leveraging global and cerebellar granule cell specific knockout of CB1 and acute pharmacological manipulations, and using the natural behavioral variability as well as experimenter-controlled manipulation of the level of locomotion using a motorized treadmill to obtain convergent evidence for the conclusions. The results provide additional support for the authors' previous work showing an influence of locomotion on eyeblink conditioning, and raise the interesting the question of what the functional role of CB1 signaling within the cerebellum might be, which could inspire more work to resolve it. More generally, this study highlights the need for caution in interpreting causal relationships between observations at the synaptic and behavioral level. If one observes that a given manipulation affects both synaptic plasticity and learning, it may be tempting to infer a causal relationship from that correlation, but the authors' findings provide a reminder that deeper analysis is required, and that the behavioral state can have widespread influence on brain function, including functions that are considered low-level or automatic.

Essential revisions:

A more nuanced interpretation of the results in the context of previous findings is merited.

1) More than one reviewer noted that the authors should include discussion of the documents effects of agonists on eyeblink conditioning, which yield different insights and conclusions about the role of cannabinoid signaling in the cerebellum than the effects of antagonists alone, which are described in this and previous studies. Cannabinoid agonists cause eyeblink conditioning deficits in humans and rats. These deficits cannot be attributed to decreased locomotor activity because humans are instructed to stay seated and rats freeze throughout the training sessions. Moreover, the deficits in rats are found with intracerebellar agonist infusions. The agonist effects combined with the current findings seem to suggest that CB1Rs within the cerebellum are not essential for cerebellar learning mechanisms under normal conditions, but cannabinoid agonists can impair cerebellar learning by artificially suppressing NT release within the cortex. The authors are encouraged to add some discussion of this issue since the effects of THC and other cannabinoid agonists on eyeblink conditioning are well documented.

2) The use of the CB1R antagonist AM251 in comparison with the global and granule cell CB1R KOs is a very good approach to address the possibility of developmental compensation in the KOs. The results from eyeblink conditioning during forced locomotion provide evidence that the results from the KOs are not due to long-term compensation. However, the data from eyeblink conditioning on the self-paced treadmill are uninterpretable, since eyeblink conditioning was profoundly impaired, even with injection of the vehicle alone. As indicated in Figure 2—figure supplement 2, the vehicle has big effects on activity as well as CRs. In fact, the vehicle and AM251 groups are impaired relative to the WT and global KO groups, even with the motorized treadmill (compare Figure 2A and B).

The reviewers appreciated that the authors showed the self-paced locomotion pharmacology results, rather than trying to sweep this limitation of the study under the rug by only showing the forced locomotion pharmacology. And after discussion, the reviewers agreed that it was not necessary to find an injection vehicle/handling procedure compatible with eyeblink conditioning during self-paced locomotion. Nevertheless, since the same vehicle and handling procedures were used in the forced locomotion experiments, it would be appropriate to include additional discussion of this limitation of the pharmacological approach, and how it influences interpretation of the results.

3) The conclusion that CB1 receptors "are dispensable for cerebellum-dependent behaviors" seems to go beyond the data showing that they are not required for eyeblink conditioning or split belt locomotor learning, under the specific experimental conditions tested. It has been reported that manipulations of specific molecules or synaptic properties can selectively effect certain forms of cerebellar learning while leaving other forms of cerebellum-dependent learning intact, hence appropriate caution is needed in extrapolating from one or two specific behavioral tasks to cerebellar learning in general. For eyeblink conditioning specifically, decades of research from Mike Mauk's lab has suggested that the cerebellar cortex is dispensable for expression of conditioned eyeblink responses, although it is indispensable for the expression of appropriately timed conditioned responses (CRs). Although the Results section indicates that "we analyzed the amplitude and timing of conditioned responses", no analysis of timing is provided beyond a general statement that the CRs peaked "around the time that the US would have been presented". Figure 1G seems to reveal a slightly earlier CR in the CB1 KO vs. WT mice, consistent with the Mauk timing hypothesis. Might an effect of blocking CB1 receptors been observed if a longer ISI was used for conditioning, requiring a more delayed blink? Also, could CB1-dependent short-term plasticity contribute to the single-trial effects on eyeblink conditioning that have been reported (Khilkovech et al., 2016)?

4) The demonstration that CB1 signaling in the cerebellum is not necessary for eyeblink conditioning raises the obvious question of what the heck the CB1 receptors in the cerebellum, and their well-documented effect on plasticity might be doing for cerebellar function. This "elephant in the room" should be at least briefly acknowledged and discussed.

5) In Figure 1F, G, more information should be provided about how the "activity-matching" was done, which the reviewers could not find anywhere in the Materials and methods or elsewhere. There are several ways in which the authors might take fuller advantage of their data to make a detailed presentation of how eyeblink conditioning depends on locomotor activity in both WT and CB1KOs mice. For example, the authors might consider labeling "top active" and "less active" individuals in Figure 1E, and dividing the CB1KOs as well as the WT into "top active" and "less active" CB1KOs in Figure 1F, G.

6) It would be informative to have various measures of behavioral states in G6 KO mice during eyeblink conditioning, as supplementary data. Compared with WT behavioral states and learning curves in Figure 1C and D, the G6 controls in Figure 3C looks more like the "less active" group because of their slower learning, especially during the first a few sessions.

7) Title: This manuscript provides evidence that altered behavioral state accounts for the effects of *reduced* cannabinoid signaling on *one* cerebellum-dependent learning task. The claim in the title is broader, and should be modified accordingly

---

## [Author Response]

Essential revisions:A more nuanced interpretation of the results in the context of previous findings is merited.1) More than one reviewer noted that the authors should include discussion of the documents effects of agonists on eyeblink conditioning, which yield different insights and conclusions about the role of cannabinoid signaling in the cerebellum than the effects of antagonists alone, which are described in this and previous studies. Cannabinoid agonists cause eyeblink conditioning deficits in humans and rats. These deficits cannot be attributed to decreased locomotor activity because humans are instructed to stay seated and rats freeze throughout the training sessions. Moreover, the deficits in rats are found with intracerebellar agonist infusions. The agonist effects combined with the current findings seem to suggest that CB1Rs within the cerebellum are not essential for cerebellar learning mechanisms under normal conditions, but cannabinoid agonists can impair cerebellar learning by artificially suppressing NT release within the cortex. The authors are encouraged to add some discussion of this issue since the effects of THC and other cannabinoid agonists on eyeblink conditioning are well documented.

We agree completely with this assessment. We have added additional information about the effects of CB1 agonists to the Introduction, and added a new paragraph to the Discussion, where we explicitly discuss our results within the context of the agonist studies. We have also revised the title to remove any implication that the behavioral state effect is the only mechanism through which cannabinoids can influence cerebellar learning (which was not our intended meaning).

2) The use of the CB1R antagonist AM251 in comparison with the global and granule cell CB1R KOs is a very good approach to address the possibility of developmental compensation in the KOs. The results from eyeblink conditioning during forced locomotion provide evidence that the results from the KOs are not due to long-term compensation. However, the data from eyeblink conditioning on the self-paced treadmill are uninterpretable, since eyeblink conditioning was profoundly impaired, even with injection of the vehicle alone. As indicated in Figure 2—figure supplement 2, the vehicle has big effects on activity as well as CRs. In fact, the vehicle and AM251 groups are impaired relative to the WT and global KO groups, even with the motorized treadmill (compare Figure 2A and B).The reviewers appreciated that the authors showed the self-paced locomotion pharmacology results, rather than trying to sweep this limitation of the study under the rug by only showing the forced locomotion pharmacology. And after discussion, the reviewers agreed that it was not necessary to find an injection vehicle/handling procedure compatible with eyeblink conditioning during self-paced locomotion. Nevertheless, since the same vehicle and handling procedures were used in the forced locomotion experiments, it would be appropriate to include additional discussion of this limitation of the pharmacological approach, and how it influences interpretation of the results.

Thank you for this comment – indeed we wanted to show this unexpected result for full disclosure, but did not deal with it extensively because we were concerned that it could be distracting. We now specifically address the limitations of this approach in the relevant sections of the Results and Materials and methods.

3) The conclusion that CB1 receptors "are dispensable for cerebellum-dependent behaviors" seems to go beyond the data showing that they are not required for eyeblink conditioning or split belt locomotor learning, under the specific experimental conditions tested. It has been reported that manipulations of specific molecules or synaptic properties can selectively effect certain forms of cerebellar learning while leaving other forms of cerebellum-dependent learning intact, hence appropriate caution is needed in extrapolating from one or two specific behavioral tasks to cerebellar learning in general. For eyeblink conditioning specifically, decades of research from Mike Mauk's lab has suggested that the cerebellar cortex is dispensable for expression of conditioned eyeblink responses, although it is indispensable for the expression of appropriately timed conditioned responses (CRs).

We have used more specific language when discussing the dispensability of CB1 receptors for the cerebellar behaviors we assessed (eyeblink conditioning, locomotor coordination, locomotor learning).

Although the Results section indicates that "we analyzed the amplitude and timing of conditioned responses", no analysis of timing is provided beyond a general statement that the CRs peaked "around the time that the US would have been presented". Figure 1G seems to reveal a slightly earlier CR in the CB1 KO vs. WT mice, consistent with the Mauk timing hypothesis. Might an effect of blocking CB1 receptors been observed if a longer ISI was used for conditioning, requiring a more delayed blink? Also, could CB1-dependent short-term plasticity contribute to the single-trial effects on eyeblink conditioning that have been reported (Khilkovech et al., 2016)?

We enthusiastically agree that given the role of CB1 receptors in short-term plasticity, the timing of learned responses and possible effects on single trial learning are the most interesting possible effects of CB1 receptor deletion – in fact those were the first two features of learning that we originally set out to test in these mice! We have added a new Figure 1—figure supplement 2, which addresses both timing and single trial effects – both of which are completely normal in CB1KOs. Thank you for this particularly insightful comment.

4) The demonstration that CB1 signaling in the cerebellum is not necessary for eyeblink conditioning raises the obvious question of what the heck the CB1 receptors in the cerebellum, and their well-documented effect on plasticity might be doing for cerebellar function. This "elephant in the room" should be at least briefly acknowledged and discussed.

We have added a discussion of this point, together with a discussion of the effects of intracerebellar CB1 agonists (Major Point 1), to a new paragraph in the Discussion.

5) In Figure 1F, G, more information should be provided about how the "activity-matching" was done, which the reviewers could not find anywhere in the Materials and methods or elsewhere. There are several ways in which the authors might take fuller advantage of their data to make a detailed presentation of how eyeblink conditioning depends on locomotor activity in both WT and CB1KOs mice. For example, the authors might consider labeling "top active" and "less active" individuals in Figure 1E, and dividing the CB1KOs as well as the WT into "top active" and "less active" CB1KOs in Figure 1F, G.

Indeed we had failed to define the activity matching and we apologize. We have modified Figure 1E to make it clear where the division was in 1F, and clarified this in the Results and figure legend as well. We also added new panels to Figure 1—figure supplement 1F and G to show that more active mice of both genotypes learned faster than less active mice of the same genotype. We note that more complete information about how activity affects learning in WT mice was reported in our 2018 paper (Albergaria et al., 2018). No differences were observed in CB1KOs.

6) It would be informative to have various measures of behavioral states in G6 KO mice during eyeblink conditioning, as supplementary data. Compared with WT behavioral states and learning curves in Figure 1C and D, the G6 controls in Figure 3C looks more like the "less active" group because of their slower learning, especially during the first a few sessions.

We have added a new Figure 3—figure supplement 1 to provide this information. G6 knockouts were not different from their littermates in any measures of behavioral state. As the reviewer notes, both controls and KOs in the Gabra6 line learned slower than WT, although this was not accounted for by differences in locomotor activity across the two lines. This is likely an effect of genetic background and it underscores why it is always critical to compare KOs to their littermate controls, as we do here.

7) Title: This manuscript provides evidence that altered behavioral state accounts for the effects of reduced cannabinoid signaling on one cerebellum-dependent learning task. The claim in the title is broader, and should be modified accordingly.

We agree and have modified the title to 1) reflect the fact that our results deal specifically with cerebellar associative learning, and 2) emphasize the effect of behavioral state, rather than implying that it is the only possible mechanism through which cannabinoids can influence cerebellar learning (in the spirit of Major Point 1).